# Random Walk Graph Neural Networks

**Giannis Nikolentzos**
École Polytechnique and AUEB
nikolentzos@aueb.gr

**Michalis Vazirgiannis**
École Polytechnique and AUEB
mvazirg@lix.polytechnique.fr

## Abstract

In recent years, graph neural networks (GNNs) have become the de facto tool for performing machine learning tasks on graphs. Most GNNs belong to the family of message passing neural networks (MPNNs). These models employ an iterative neighborhood aggregation scheme to update vertex representations. Then, to compute vector representations of graphs, they aggregate the representations of the vertices using some permutation invariant function. One would expect the hidden layers of a GNN to be composed of parameters that take the form of graphs. However, this is not the case for MPNNs since their update procedure is parameterized by fully-connected layers. In this paper, we propose a more intuitive and transparent architecture for graph-structured data, so-called Random Walk Graph Neural Network (RWNN). The first layer of the model consists of a number of trainable "hidden graphs" which are compared against the input graphs using a random walk kernel to produce graph representations. These representations are then passed on to a fully-connected neural network which produces the output. The employed random walk kernel is differentiable, and therefore, the proposed model is end-to-end trainable. We demonstrate the model's transparency on synthetic datasets. Furthermore, we empirically evaluate the model on graph classification datasets and show that it achieves competitive performance.

## 1 Introduction

In recent years, graphs have become a very useful abstraction for representing a wide variety of real-world datasets. Graphs are ubiquitous in several application domains, such as in social networks, in bioinformatics, and in information networks. Due to this abundance of graph-structured data, machine learning on graphs has recently emerged as a very important task with applications ranging from drug design [18] to modeling physical systems [3].

In the past years, graph neural networks (GNNs) have attracted considerable attention in the machine learning community. These models offer a powerful tool for performing machine learning on graphs. Although numerous GNN variants have been recently proposed [35, 24, 23, 48, 46, 44], most of them share the same basic idea, and can be reformulated into a single common framework, so-called message passing neural networks (MPNNs) [13]. These models employ a message passing procedure to aggregate local information of vertices. For graph-related tasks, MPNNs usually apply some permutation invariant readout function to the vertex representations to produce a representation for the entire graph. Common readout functions treat each graph as a set of vertex representations, thus ignoring the interactions between the vertices. These interactions are implicitly encoded into the learned vertex representations produced by the message passing procedure. However, due to the lack of transparency brought by the combination of graph structure with feature information, it is not clear whether the information of interest is encoded into these representations.

Before the advent of deep learning, graph kernels had established themselves as the standard approach for performing machine learning tasks on graphs [33, 22]. A graph kernel is a symmetric positive semidefinite function defined on the space of graphs. These methods enable the application of kernel

methods such as the SVM classifier to graphs, and have achieved remarkable results in several classification tasks. However, nowadays, they have been largely overshadowed by GNNs. This is mainly due to the complexity of kernel methods, but also because data representation (produced by graph kernels) and learning (performed by SVM) are independent from each other. On the other hand, neural network models can learn representations that are useful for the task at hand. In contrast to graph kernels, which directly compare graphs to each other, MPNNs first transform graphs into vectors by aggregating the representations of their vertices, and then some function is applied to these graph representations (i. e., modeled by a multi-layer perceptron). Furthermore, it is usually hard to interpret and understand what these models have learned. Ideally, we would like to have a model that applies directly some function to the input graphs without first transforming them into vectors.

In this paper, we propose such an architecture, called Random Walk Graph Neural Network (RWNN). The model contains a number of trainable "hidden graphs", and it compares the input graphs against these graphs using a random walk kernel. The kernel values are then passed on to a fully-connected neural network which produces the output. The employed random walk kernel is differentiable, and we can thus update the "hidden graphs" during training with backpropagation. Hence, the proposed neural network is end-to-end trainable. Furthermore, it delivers the "best of both worlds" from graph kernels and neural networks, i. e., it retains the flexibility of kernel methods which can be easily applied to structured data (e. g., graphs), and also learns task-dependent features without the need for feature engineering. We compare the performance of the proposed model to state-of-the-art graph kernels and recently-proposed neural architectures on several benchmark datasets for graph classification. Results show that our model matches or outperforms competing methods. Our main contributions are summarized as follows:

- We propose a novel neural network model, Random Walk Graph Neural Network, which employs a random walk kernel to produce graph representations. Importantly, the model is highly interpretable since it contains a set of trainable graphs.
- We develop an efficient computation scheme to reduce the time and space complexity of the proposed model.
- We demonstrate the model's high transparency on synthetic datasets, and evaluate its performance on several graph classification datasets where it achieves performance comparable to state-of-the-art GNNs and graph kernels.

The rest of this paper is organized as follows. Section 2 provides an overview of the related work. Section 3 introduces some preliminary concepts. Section 4 provides a detailed description of the proposed model. Section 5 evaluates the proposed model both in terms of its transparency and of its performance. Finally, Section 6 concludes.

## 2 Related Work

In the past years, graph kernels have served as the dominant tool for graph classification. Graph kernels are positive semidefinite kernel functions which enable the applicability of the whole family of kernel methods to the domain of graphs. Most graph kernels are instances of the R-convolution framework and compare different types of substructures in the input graphs. Such substructures include random walks [17, 12, 25, 43, 42], shortest paths [5], subtrees [34], graphlets [37], etc. Johansson *et al.* introduced two graph kernels that compare subgraphs based on the Lovász number and the corresponding orthonormal representations [16], while Kondor and Pan developed the multiscale Laplacian graph kernel which captures similarity at different granularity levels by building a hierarchy of nested subgraphs [20]. The Weisfeiler-Lehman framework operates on top of existing kernels and uses a relabeling procedure that is based on the Weisfeiler-Lehman isomorphism test [36]. Recently, the family of assigmnent kernels has gained some attention. For instance, Nikolentzos *et al.* proposed an assignment kernel for graphs that capitalizes on the well-known pyramid match kernel [32] , while Kriege *et al.* presented a framework for building valid optimal assignment kernels, and they derived three graph kernels from that framework [21].

The concept of graph neural networks (GNNs) has been around for several years [40, 26, 35]. However, these models had received relatively little attention until recently, when Li *et al.* modified the model proposed in [35] to use modern practices around recurrent neural networks and optimization techniques [24]. Some other models that are based on spectral properties of graphs were also proposed at that time. Bruna *et al.* generalized the convolution operator in the domain of graphs using the

eigenvectors of the Laplacian matrix [6], while Defferrard *et al.* proposed a more efficient model which uses Chebyshev polynomials up to order $K - 1$ to represent the spectral filters [9]. Later, it became clear that all these models are special cases of a simple message-passing framework (MPNNs) [13]. Most of the recently-proposed GNNs fit into this framework [23, 48, 44, 28]. Specifically, MPNNs employ a message passing procedure, where each vertex updates its feature vector by aggregating the feature vectors of its neighbors. After $k$ iterations of the message passing procedure, each vertex obtains a feature vector which captures the structural information within its $k$-hop neighborhood. MPNNs then compute a feature vector for the entire graph using some permutation invariant readout function such as summing the feature vectors of all the vertices of the graph. Some models employ advanced pooling strategies for learning hierarchical graph representations [41, 46]. The family of MPNNs is closely related to the Weisfeiler-Lehman subtree kernel (WL) [36]. Specifically, these models generalize the relabeling procedure of the WL kernel to the case where vertices are associated with continuous feature vectors. Standard MPNNs have been shown to be at most as powerful as the WL kernel in distinguishing non-isomorphic graphs [44, 27]. There are models which are not members of the MPNN family. For instance, Niepert *et al.* proposed a model that extracts neighborhood subgraphs for a subset of vertices, imposes an ordering on the vertices of each subgraph, and then feeds the adjacency and feature matrices of these sugraphs to a convolutional neural network [30]. The work closest to ours is the one reported in [23], where the authors propose a class of GNNs whose generated representations are associated with either the random walk kernel or the WL kernel. Specifically, the outputs of these models live in the reproducing kernel Hilbert space (RKHS) of these kernels. However, the kernels are computed between the input graphs and the row vectors from the parameter matrices. Unfortunately, these vectors cannot always be mapped to graph structures. Our model differs from theirs in that its parameters correspond to the adjacency and feature matrices of graphs. In [8], Chen *et al.* generate finite-dimensional vertex representations using the Nyström method to approximate a kernel that compares a set of local patterns centered at vertices. These representations can be learned without supervision by extracting a set of anchor points, or can be modeled as parameters of a neural network and be learned end-to-end. Other works that merge neural networks and graph kernels include [29, 31, 10] and [1]. In [29], Navarin *et al.* use graph kernels to pre-train GNNs. In [31], Nikolentzos *et al.* use graph kernels to extract features that are then passed on to convolutional neural networks, while in [10], Du *et al.* follow the opposite direction and propose a new graph kernel which corresponds to infinitely wide multi-layer GNNs trained by gradient descent. Finally, Al-Rfou *et al.* propose in [1] an unsupervised method for learning graph representations by comparing the input graphs against a set of source graphs. However, the source graphs are fixed and not trainable. Our work is also related to explainability techniques for GNNs [45, 2, 47]. We should stress, however, that the main focus of the proposed model is not on providing interpretable explanations for its predictions, but these explanations come as a byproduct of the learning process.

## 3 Preliminaries

Before continuing with our contribution, we begin by introducing the problem which we address in this paper and some key notation for graphs which will be used later.

In this paper, we focus on the graph classification problem. Formally, given a set of graphs $\{G_1, \ldots, G_N\} \subseteq \mathcal{G}$ and their class labels $\{y_1, \ldots, y_N\} \subseteq \mathcal{Y}$, we aim to learn a feature vector $\mathbf{h}_G$ that will allow us to predict the class label of graph $G$, i.e., $y_G = f(\mathbf{h}_G)$ where $f$ is some function. Note that the proposed model can also deal with other graph-related tasks such as the graph regression problem.

Let $G = (V, E)$ be an undirected graph, where $V$ is the vertex set and $E$ is the edge set. We will denote by $n$ the number of vertices and by $m$ the number of edges. The adjacency matrix $\mathbf{A} \in \mathbb{R}^{n \times n}$ is a symmetric (typically sparse) matrix used to encode edge information in a graph. The elements of the $i^{th}$ row and $j^{th}$ column is equal to 1 if there is an edge between $v_i$ and $v_j$, and 0 otherwise. For node-attributed graphs, every node in the graph is associated with a feature vector. We use $\mathbf{X} \in \mathbb{R}^{n \times d}$ to denote the node features where $d$ is the feature dimensionality. The feature of a given node $v_i$ corresponds to the $i^{th}$ row of $\mathbf{X}$. Given two graphs $G = (V, E)$ and $G' = (V', E')$, their direct product $G_\times = (V_\times, E_\times)$ is a graph with vertex set $V_\times = \{(v, v') : v \in V \wedge v' \in V'\}$ and edge set $E_\times = \{\{(v, v'), (u, u')\} : \{v, u\} \in E \wedge \{v', u'\} \in E'\}$. From the above definition of the direct product, it is clear that $G_\times$ is a graph over pairs of vertices from $G$ and $G'$, and two vertices in $G_\times$ are neighbors if and only if the corresponding vertices in $G$ and $G'$ are both neighbors.

# 4 Random Walk GNNs

One of the main challenges when designing neural networks for graphs is how to deal with permutation invariance. Any permutation of the vertices of a graph gives rise to a structurally identical graph. Therefore, the output of a GNN is necessary to be the same regardless of the ordering of the graph's vertices. A graph is commonly represented by its adjacency matrix and possibly by a matrix that contains the attributes of its vertices. Given the adjacency matrix $\mathbf{A}$ of a graph $G$, the model is necessary to produce the same output for all matrices $\mathbf{P}\,\mathbf{A}\,\mathbf{P}^\top$ where $\mathbf{P} \in \Pi$ and $\Pi$ denotes the set of $n \times n$ permutation matrices. Most MPNNs achieve that by applying some permutation invariant function to the vertex representations (e. g., sum, max, mean operators). Other GNNs, achieve invariance by imposing an ordering on the graph's vertices usually using some heuristic. In this paper, we propose a different approach for obtaining graph representations. We capitalize on well-established concepts from graph kernels and we employ functions for comparing graphs that are invariant to permutations of the vertices of their inputs.

The proposed RWNN model compares the input graphs against a number of "hidden graphs", i. e., graphs whose adjacency and attribute matrices are trainable. Specifically, the proposed model contains $N$ "hidden graphs" $G_1, G_2, \ldots, G_N$. The graphs may differ from each other in terms of size (i. e., number of vertices). These graphs may be either unlabeled or their vertices may be annotated with continuous multi-dimensional features. Furthermore, as mentioned above, the structure and vertex attributes (if any) of these "hidden graphs" are trainable, i. e., the adjacency matrix of a "hidden graph" $G_i$ of size $n$ is described by a trainable matrix $\mathbf{W}_i \in \mathbb{R}^{n \times n}$, while the vertex attributes are contained in the rows of another trainable matrix $\mathbf{Z}_i \in \mathbb{R}^{n \times d}$. Note that the "hidden graphs" can be weighted directed or undirected graphs with or without self-loops. In our implementation, we constraint them to be undirected graphs without self-loops ($n(n-1)/2$ trainable parameters in total). Since the structure of the "hidden graphs" is adapted to the task at hand, the proposed model is highly interpretable. By visualizing the structure of the "hidden graphs" at the end of the training phase, we can gain a more intuitive understanding of the considered problem. These graphs are expected to learn structures which allow the model to distinguish between available classes.

In contrast to existing approaches, in this paper, to map input graphs to vectors, we compare them against the model's $N$ "hidden graphs". It should be noted that the graph comparison algorithm needs to be differentiable in order for the network to be end-to-end trainabale. Otherwise, the structure and attributes of the "hidden graphs" cannot be learned during training with backpropagation. Even though there exist hundreds of algorithms for comparing graphs to each other, unfortunately, the majority of these algorithms are not differentiable. For non-differentiable graph comparison algorithms, the "hidden graphs" could be held constant during training. However, that would limit a lot the expressive power of the model. In this paper, we propose a differentiable function for comparing graphs to each other which belongs to the family of the random walk kernels, perhaps the most well-studied family of graph kernels. Generally speaking, the random walk kernels quantify the similarity of two graphs based on the number of common walks in the two graphs [17, 12, 25, 43, 42, 49]. Among the numerous variations of the random walk kernel, in this paper, we focus on the $P$-step random walk kernel which compares random walks up to length $P$ in the two graphs.

Performing a random walk on the direct product $G_\times$ (introduced in Section 3) of two graphs $G$ and $G'$ is equivalent to performing a simultaneous random walk on the two graphs $G$ and $G'$. We denote by $\mathbf{A}_\times$ the adjacency matrix of $G_\times$. As mentioned above, random walk kernels count all pairs of matching walks on $G$ and $G'$. If we assume a uniform distribution for the starting and stopping probabilities over the vertices of $G$ and $G'$, the number of matching walks can be obtained through the adjacency matrix $\mathbf{A}_\times$ of the product graph $G_\times$ [43]. Given some $P \in \mathbb{N}$, the $P$-step random walk kernel between two graphs $G$ and $G'$ is defined as:

$$k(G, G') = \sum_{i=1}^{|V_\times|} \sum_{j=1}^{|V_\times|} \left[ \sum_{p=0}^{P} \lambda_p \mathbf{A}_\times^p \right]_{ij} \tag{1}$$

with a sequence of positive, real-valued weights $\lambda_0, \lambda_1, \ldots, \lambda_P$. The proposed RWNN model computes a slight variant of the $P$-step random walk kernel which calculates the number of common walks of length exactly $p$ between two graphs $G$ and $G'$:

$$k^{(p)}(G, G') = \sum_{i=1}^{|V_\times|} \sum_{j=1}^{|V_\times|} \left[ \mathbf{A}_\times^p \right]_{ij} \tag{2}$$

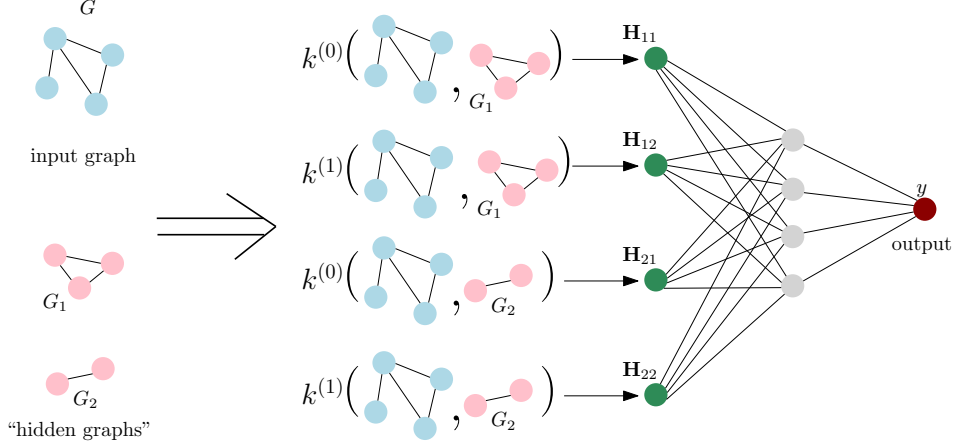

Figure 1: Overview of the proposed RWNN model (biases are omitted for clarity).

Then, for each $p \in \{0, 1, \ldots, P\}$, we obtain a different kernel value. These kernel values can be thought of as features of the input graph. Therefore, given the two sets $\mathcal{P} = \{0, 1, \ldots, P\}$ and $\mathcal{G}_h = \{G_1, G_2, \ldots, G_N\}$ where $G_1, G_2, \ldots, G_N$ denote the $N$ "hidden graphs", we can compute one feature for each element of the Cartesian product $\mathcal{P} \times \mathcal{G}_h$. For each input graph $G$, we can thus build a matrix $\mathbf{H} \in \mathbb{R}^{N \times P}$ where $\mathbf{H}_{ij} = k^{(j-1)}(G, G_i)$. Then, matrix $\mathbf{H}$ is flattened and fed into a fully-connected neural network to produce the output. The proposed architecture is illustrated in Figure 1.

Unfortunately, the architecture that was presented above cannot handle graphs whose vertices are annotated with real-valued multi-dimensional vertex attributes. We next generalize the above function to graphs that contain such vertex attributes. Specifically, let $\mathbf{X} \in \mathbb{R}^{n \times d}$ denote the matrix that contains the vertex attributes of the input graph $G$. We also annotate the vertices of the "hidden graphs" with vectors of the same dimensionality. For each "hidden graph" $G_i$, these vectors are represented by the rows of a trainable matrix $\mathbf{Z}_i \in \mathbb{R}^{c \times d}$ where $c$ is the number of vertices of $G_i$. Then, let $\mathbf{S} = \mathbf{Z}_i \mathbf{X}^\top$ where $\mathbf{S} \in \mathbb{R}^{c \times n}$. The $(i, j)^{th}$ element (with a slight abuse of notation since it is clear from the context) of matrix $\mathbf{S}$ is equal to the inner product between the attributes of the $i^{th}$ vertex of the "hidden graph" and the $j^{th}$ vertex of the input graph $G$. Roughly speaking, this matrix encodes the similarity between the attributes of the vertices of the two graphs. Let vec denote the vectorization operator which transforms a matrix into a vector by stacking the columns of the matrix one after another (see the supplementary material for the exact definition). We can then apply this operator to matrix $\mathbf{S}$ to obtain $\mathbf{s} = \text{vec}(\mathbf{S})$ where $\mathbf{s} \in \mathbb{R}^{nc}$. Then, we can compute the kernel that counts the number of common walks of length exactly $p$ between the two graphs as follows:

$$k^{(p)}(G, G') = \sum_{i=1}^{|V_\times|} \sum_{j=1}^{|V_\times|} \mathbf{s}_i \mathbf{s}_j \left[\mathbf{A}_\times^p\right]_{ij} = \sum_{i=1}^{|V_\times|} \sum_{j=1}^{|V_\times|} \left[(\mathbf{s}\,\mathbf{s}^\top) \odot \mathbf{A}_\times^p\right]_{ij} = \mathbf{s}^\top \mathbf{A}_\times^p \mathbf{s} \qquad (3)$$

where $\odot$ denotes the element-wise product between two matrices. Note that the $(i, j)^{th}$ element of matrix $\mathbf{A}_\times^p$ is equal to the number of walks of length $p$ between the $i^{th}$ and $j^{th}$ vertex of $G_\times$. Each vertex of $G_\times$ corresponds to a pair of vertices, one from the input graph $G$ and one from the "hidden graph" $G_i$. We can assign a real value to each vertex of $G_\times$ that quantifies the similarity between the attributes of the two vertices it represents. These values are contained in vector $\mathbf{s}$. Note that if $\mathbf{s}$ is the $nc$-dimensional all-ones vector, then Equation (3) becomes equal to Equation (2).

## 4.1 Implementation Details

In what follows, we give more details about the implementation of the proposed RWNN model. Importantly, we present the computation scheme that we employed to reduce the time and space complexity of the proposed architecture.

As mentioned above, in our implementation, the "hidden graphs" correspond to undirected graphs without self-loops. The adjacency matrix of a "hidden graph" with $n$ vertices is thus represented

by $n(n-1)/2$ trainable parameters. Since these graphs are trainable, the elements of their adjacency matrices may take negative values. Although this is not a problem in general, to increase the interpretability of the model, we restrict the edge weights to nonnegative real values by applying the ReLU function to the adjacency matrices of the "hidden graphs". It is known that if $\mathbf{A}$ and $\mathbf{A}'$ are the respective adjacency matrices of two graphs $G$ and $G'$, then we have that the adjacency matrix $\mathbf{A}_\times$ of the direct product $G_\times$ is equal to $\mathbf{A}_\times = \mathbf{A} \otimes \mathbf{A}'$ where $\otimes$ denotes the Kronecker product between two matrices (see the supplementary material for the exact definition) [43]. Therefore, in our setting, if $\mathbf{A}$ and $\text{RELU}(\mathbf{W}_i)$ are the adjacency matrices of the input graph $G$ and a "hidden graph" $G_i$, then the adjacency matrix of the direct product is equal to $\mathbf{A}_\times = \mathbf{A} \otimes \text{RELU}(\mathbf{W}_i)$.

We will next show that in order to evaluate the kernel defined in Equation (3), it is not necessary to explicitly compute the matrix $\mathbf{A}_\times$ and its powers. The Kronecker product and vec operator are linked by the well-known property [4](Proposition 7.1.9):

$$\text{vec}(\mathbf{ABC}) = (\mathbf{C}^\top \otimes \mathbf{A})\text{vec}(\mathbf{B}) \qquad (4)$$

Let $\text{vec}^{-1}$ denote the inverse vectorization operator which transforms a vector into matrix (see the supplementary material for the exact definition). Given matrices $\mathbf{A}, \text{vec}^{-1}(\mathbf{b})$ and $\mathbf{C}$, Equation (4) can be written as:

$$\text{vec}\big(\mathbf{A}\,\text{vec}^{-1}(\mathbf{b})\,\mathbf{C}\big) = (\mathbf{C}^\top \otimes \mathbf{A})\,\text{vec}\big(\text{vec}^{-1}(\mathbf{b})\big) = (\mathbf{C}^\top \otimes \mathbf{A})\mathbf{b}$$

From Equation (3) and based on the above Equation, we have:

$$
\begin{aligned}
k^{(1)}(G, G_i) = \mathbf{s}^\top \mathbf{A}_\times \mathbf{s} &= \mathbf{s}^\top \big(\mathbf{A} \otimes \text{RELU}(\mathbf{W}_i)\big)\mathbf{s} = \mathbf{s}^\top\big(\mathbf{A}^\top \otimes \text{RELU}(\mathbf{W}_i)\big)\,\mathbf{s} \\
&= \mathbf{s}^\top \text{vec}\big(\text{RELU}(\mathbf{W}_i)\,\text{vec}^{-1}(\mathbf{s})\,\mathbf{A}\big)
\end{aligned}
$$

The third equality holds beacuse we have assumed that the input graphs are undirected. Note that the above result holds only for $p = 1$. To generalize the result, it is necessary to show that $\mathbf{A}_\times^p = \mathbf{A}_1^p \otimes \mathbf{A}_2^p$ holds for all $p \in \mathbb{N}$.

**Proposition 1.** *Let $\mathbf{A}_1 \in \mathbb{R}^{n \times n}$ and $\mathbf{A}_2 \in \mathbb{R}^{m \times m}$ be two real matrices such that $\mathbf{A}_\times = \mathbf{A}_1 \otimes \mathbf{A}_2$. Then, for any $p \in \mathbb{N}$, we have that $\mathbf{A}_\times^p = (\mathbf{A}_1 \otimes \mathbf{A}_2)^p = \mathbf{A}_1^p \otimes \mathbf{A}_2^p$.*

Based on the above result (the proof is left to the supplementary material), we have that:

$$
\begin{aligned}
k^{(p)}(G, G_i) &= \mathbf{s}^\top \text{vec}\big(\text{RELU}(\mathbf{W}_i)^p\,\text{vec}^{-1}(\mathbf{s})\,\mathbf{A}^p\big) = \mathbf{s}^\top \text{vec}\big(\text{RELU}(\mathbf{W}_i)^p\,\mathbf{S}\,\mathbf{A}^p\big) \\
&= \mathbf{s}^\top \text{vec}\big(\text{RELU}(\mathbf{W}_i)^p\,\mathbf{Z}_i\,\mathbf{X}^\top\,\mathbf{A}^p\big) = \text{vec}(\mathbf{S})^\top \text{vec}\big(\text{RELU}(\mathbf{W}_i)^p\,\mathbf{Z}_i\,\mathbf{X}^\top\,\mathbf{A}^p\big) \\
&= \text{vec}(\mathbf{Z}\,\mathbf{X}^\top)^\top \text{vec}\big(\text{RELU}(\mathbf{W}_i)^p\,\mathbf{Z}_i\,\mathbf{X}^\top\,\mathbf{A}^p\big) \\
&= \sum_{j=1}^{r} \sum_{k=1}^{n} \big[\mathbf{Z}_i\,\mathbf{X}^\top \odot \big(\text{RELU}(\mathbf{W}_i)^p\,\mathbf{Z}_i\,\mathbf{X}^\top\,\mathbf{A}^p\big)\big]_{jk}
\end{aligned}
$$

Note that we can compute $k^{(p)}(G, G_i)$ without calculating $\mathbf{A}^p$ and $\text{RELU}(\mathbf{W}_i)^p$. We can calculate $\text{RELU}(\mathbf{W}_i)^p\,\mathbf{Z}_i$ and $\mathbf{X}^\top\,\mathbf{A}^p$ with right-to-left and left-to-right multiplications, respectively. For instance, for $p = 3$, we can compute $\mathbf{X}^\top\,\mathbf{A}^3$ as $\big((\mathbf{X}^\top\,\mathbf{A})\,\mathbf{A}\big)\,\mathbf{A}$. Since we store $\mathbf{A}$ as a sparse matrix with $m$ non-zero entries, an efficient implementation of our model takes $\mathcal{O}\big(Pd(Nc(c+n) + m)\big)$ computational time, where $P$ is the size of the longest walks, $d$ is the dimensionality of the vertex attributes, $N$ is the number of "hidden graphs", and $c$ is the size of each "hidden graph". Under the realistic assumptions of $P \ll m$ and $d \ll m$, running the model takes $\mathcal{O}\big(Nc(c+n) + m\big)$ computational time.

If the vertex attributes are very high-dimensional, we can transform them using a 1-layer perceptron, i. e., $\tilde{\mathbf{X}} = f(\mathbf{X}\,\mathbf{W} + \mathbf{b})$ where $\mathbf{W} \in \mathbb{R}^{d \times \tilde{d}}$ and $\mathbf{b} \in \mathbb{R}^{\tilde{d}}$ is a weight matrix and bias vector, respectively, and $f$ is a non-linear activation function. Alternatively, we can even use an MPNN architecture to update these vertex representations, i. e., $\tilde{\mathbf{X}} = \text{MPNN}(\mathbf{A}, \mathbf{X})$.

## 5 Experimental Evaluation

In this Section, we empirically evaluate the proposed architecture on synthetic and real-world datasets, and we compare it to several baseline methods.

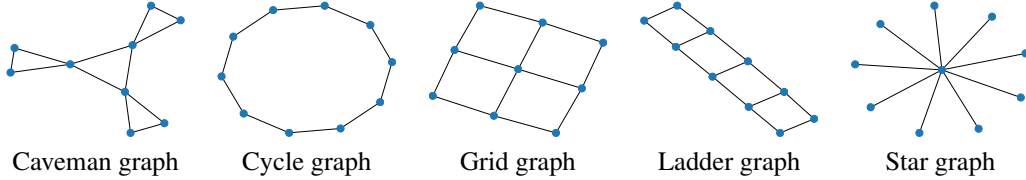

| Caveman graph | Cycle graph | Grid graph | Ladder graph | Star graph |

Figure 2: Structures planted into synthetic graphs.

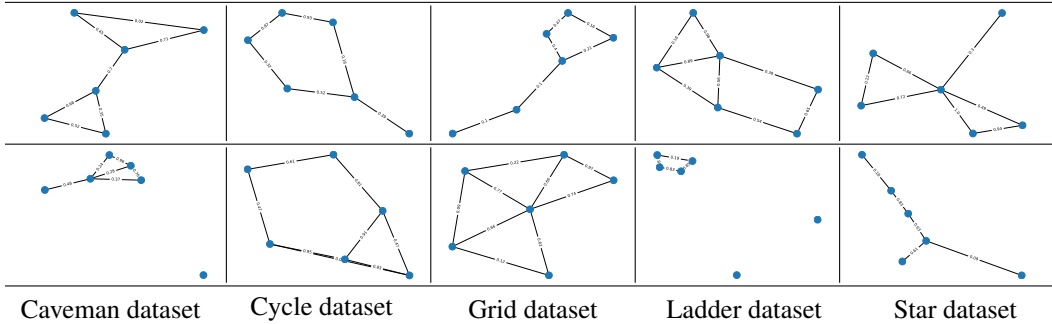

| Caveman dataset | Cycle dataset | Grid dataset | Ladder dataset | Star dataset |

Figure 3: Examples of "hidden graphs" extracted from the proposed model for the 5 synthetic datasets.

## 5.1 Synthetic Datasets

As previously mentioned, the proposed model is highly interpretable since we can visualize the learned "hidden graphs", and possibly discover the explanatory factors of variation behind the data. The aim of these experiments is to qualitatively investigate what types of graph structures are learned by RWNN.

**Datasets.** We created 5 binary classification datasets, each featuring 1000 synthetic graphs generated as follows. First, we generate an Erdös-Rényi graph with number of vertices sampled from $\mathbb{Z} \cap [100, 200]$ with uniform probability, and edge probability equal to $0.1$. From this graph, we obtain a positive and a negative sample by addding to the graph a dataset specific motif (the 5 motifs that we considered are shown in Figure 2) and an Erdös-Rényi graph with the same number of nodes and edges to the motif, respectively. The vertices of the motifs/Erdös-Rényi graphs are connected to the vertices of the previously-generated Erdös-Rényi graph with probability $0.05$.

**Experimental Setup.** We randomly split each dataset into a $90\%/10\%$ training/validation set. We set the number of epochs to $50$ and the batch size to $32$. We use the Adam optimizer with learning rate $0.001$. We set the number of "hidden graphs" to $8$ and their size equal to $6$ vertices. We set the hyperparameter $P$ of the random walks to $3$, and we use no vertex attributes. We store the model that achieved the best validation accuracy into disk. At the end of training, the model is retrieved from the disk, the adjacency matrices of the "hidden graphs" are extracted (i. e., $\text{RELU}(\mathbf{W}_i)$ matrices) and visualized.

**Results.** Figure 3 illustrates two "hidden graphs" for each of the 5 datasets. Interestingly, the "hidden graphs" and their corresponding motifs share some similar properties. For instance, the caveman graph consists of $4$ triangles, and similarly, each of the two "hidden graphs" contains $2$ triangles. The "hidden graphs" extracted from the cycle dataset both contain some long cycles, while in the "hidden graphs" extracted from the star dataset, there are some vertices with higher degree centrality than the others. Finally, the graphs of the grid dataset and the ladder dataset contain cycles of length $4$, the main building block of these two motifs. Note that the "hidden graphs" are weighted graphs. Original resolutions of the illustrations of these graphs as well as some more examples of "hidden graphs" are provided in the supplementary material.

## 5.2 Real-World Datasets

We next evaluate the proposed RWNN model on standard graph classification datasets.

Table 1: Classification accuracy ($\pm$ standard deviation) of the proposed model and the baselines on the 5 chemo/bio-informatics and on the 5 social interaction benchmark datasets. OOR means Out of Resources, either time (>72 hours for a single training) or GPU memory. Best performance per dataset in **bold**, among the neural network architectures underlined.

| | MUTAG | D&D | NCI1 | PROTEINS | ENZYMES |
|---|---|---|---|---|---|
| SP | 80.2 ($\pm$ 6.5) | **78.1** ($\pm$ 4.1) | 72.7 ($\pm$ 1.4) | **75.3** ($\pm$ 3.8) | 38.3 ($\pm$ 8.0) |
| GR | 80.8 ($\pm$ 6.4) | 75.4 ($\pm$ 3.4) | 61.8 ($\pm$ 1.7) | 71.6 ($\pm$ 3.1) | 25.1 ($\pm$ 4.4) |
| WL | 84.6 ($\pm$ 8.3) | **78.1** ($\pm$ 2.4) | **84.8** ($\pm$ 2.5) | 73.8 ($\pm$ 4.4) | 50.3 ($\pm$ 5.7) |
| DGCNN | 84.0 ($\pm$ 6.7) | 76.6 ($\pm$ 4.3) | 76.4 ($\pm$ 1.7) | 72.9 ($\pm$ 3.5) | 38.9 ($\pm$ 5.7) |
| DiffPool | 79.8 ($\pm$ 7.1) | 75.0 ($\pm$ 3.5) | 76.9 ($\pm$ 1.9) | 73.7 ($\pm$ 3.5) | 59.5 ($\pm$ 5.6) |
| ECC | 75.4 ($\pm$ 6.2) | 72.6 ($\pm$ 4.1) | 76.2 ($\pm$ 1.4) | 72.3 ($\pm$ 3.4) | 29.5 ($\pm$ 8.2) |
| GIN | 84.7 ($\pm$ 6.7) | 75.3 ($\pm$ 2.9) | 80.0 ($\pm$ 1.4) | 73.3 ($\pm$ 4.0) | **59.6** ($\pm$ 4.5) |
| GraphSAGE | 83.6 ($\pm$ 9.6) | 72.9 ($\pm$ 2.0) | 76.0 ($\pm$ 1.8) | 73.0 ($\pm$ 4.5) | 58.2 ($\pm$ 6.0) |
| 1-step RWNN | **89.2** ($\pm$ 4.3) | 77.6 ($\pm$ 4.7) | 71.4 ($\pm$ 1.8) | 74.7 ($\pm$ 3.3) | 56.7 ($\pm$ 5.2) |
| 2-step RWNN | 88.1 ($\pm$ 4.8) | 76.9 ($\pm$ 4.6) | 73.0 ($\pm$ 2.0) | 74.1 ($\pm$ 2.8) | 57.4 ($\pm$ 4.9) |
| 3-step RWNN | 88.6 ($\pm$ 4.1) | 77.4 ($\pm$ 4.9) | 73.9 ($\pm$ 1.3) | 74.3 ($\pm$ 3.3) | 57.6 ($\pm$ 6.3) |

| | IMDB BINARY | IMDB MULTI | REDDIT BINARY | REDDIT MULTI-5K | COLLAB |
|---|---|---|---|---|---|
| SP | 57.7 ($\pm$ 4.1) | 39.8 ($\pm$ 3.7) | 89.0 ($\pm$ 1.0) | 51.1 ($\pm$ 2.2) | **79.9** ($\pm$ 2.7) |
| GR | 63.3 ($\pm$ 2.7) | 39.6 ($\pm$ 3.0) | 76.6 ($\pm$ 3.3) | 38.1 ($\pm$ 2.3) | 71.1 ($\pm$ 1.4) |
| WL | **72.8** ($\pm$ 4.5) | **51.2** ($\pm$ 6.5) | 74.9 ($\pm$ 1.8) | 49.6 ($\pm$ 2.0) | 78.0 ($\pm$ 2.0) |
| DGCNN | 69.2 ($\pm$ 3.0) | 45.6 ($\pm$ 3.4) | 87.8 ($\pm$ 2.5) | 49.2 ($\pm$ 1.2) | 71.2 ($\pm$ 1.9) |
| DiffPool | 68.4 ($\pm$ 3.3) | 45.6 ($\pm$ 3.4) | 89.1 ($\pm$ 1.6) | 53.8 ($\pm$ 1.4) | 68.9 ($\pm$ 2.0) |
| ECC | 67.7 ($\pm$ 2.8) | 43.5 ($\pm$ 3.1) | OOR | OOR | OOR |
| GIN | 71.2 ($\pm$ 3.9) | 48.5 ($\pm$ 3.3) | 89.9 ($\pm$ 1.9) | **56.1** ($\pm$ 1.7) | 75.6 ($\pm$ 2.3) |
| GraphSAGE | 68.8 ($\pm$ 4.5) | 47.6 ($\pm$ 3.5) | 84.3 ($\pm$ 1.9) | 50.0 ($\pm$ 1.3) | 73.9 ($\pm$ 1.7) |
| 1-step RWNN | 70.8 ($\pm$ 4.8) | 47.8 ($\pm$ 3.8) | **90.4** ($\pm$ 1.9) | 51.7 ($\pm$ 1.5) | 71.7 ($\pm$ 2.1) |
| 2-step RWNN | 70.6 ($\pm$ 4.4) | 48.8 ($\pm$ 2.9) | 90.3 ($\pm$ 1.8) | 51.7 ($\pm$ 1.4) | 71.3 ($\pm$ 2.1) |
| 3-step RWNN | 70.7 ($\pm$ 3.9) | 47.8 ($\pm$ 3.5) | 89.7 ($\pm$ 1.2) | 53.4 ($\pm$ 1.6) | 71.9 ($\pm$ 2.5) |

**Datasets.** We evaluated the proposed model on 10 publicly available graph classification datasets including 5 bio/chemo-informatics datasets: MUTAG, D&D, NCI1, PROTEINS, ENZYMES, and 5 social interaction datasets: IMDB-BINARY, IMDB-MULTI, REDDIT-BINARY, REDDIT-MULTI-5K, COLLAB [19]. More details about the datasets are given in the supplementary material.

**Experimental Setup.** We compare the proposed model against the following three graph kernels: (1) graphlet kernel (GR) [37], (2) shortest path kernel (SP) [5], and (3) Weisfeiler-Lehman subtree kernel (WL) [36]. We use the implementations of the kernels contained in the GraKeL library [38]. We also compare our model against the following five state-of-the-art MPNNs: (1) DGCNN [48], (2) DiffPool [46], (3) ECC [39], (4) GIN [44], and (5) GraphSAGE [14]. To evaluate the different methods, we employ the framework proposed in [11]. Therefore, we perform 10-fold cross-validation to obtain an estimate of the generalization performance of each method, while within each fold a model is selected based on a $90\%/10\%$ split of the training set. We use exactly the same splits as in [11], hence, for the common datasets, we use the results reported in [11]. For the remaining datasets, we use the code provided by the authors of [11] to evaluate the five MPNNs.

For the graph kernels, to perform classification, we employed the LIBSVM implementation of the $C$-Support Vector Machine (SVM) classifier [7], and we optimize its parameter $C$ within each fold. We also chose the number of iterations of the WL kernel from $h = \{4, 5, 6, 7\}$. For the GR kernel, we set the number of graphlets to be sampled from each graph equal to $500$.

For the proposed RWNN, we provide results for three different instances: 1-step RWNN, 2-step RWNN, and 3-step RWNN for $P = 1$, $P = 2$, and $P = 3$, respectively. Note that for $P = 1$ (random walks of length up to 1), the model takes into account only walks of length 1 (i. e., edges) between the vertices. For all instances and all datasets, we set the batch size to $64$ and the number of epochs to $500$. We use the Adam optimizer with initial learning rate $0.01$ and decay the learning rate by $0.5$ every 50 epochs. We use a 1-layer perceptron to transform the vertex attributes. Batch normalization [15] is applied on the generated graph representations (i. e., matrix $\mathbf{H}$). The hyper-parameters we

tune for each dataset are: (1) the number of hidden graphs $\in \{8, 16\}$, (2) the number of vertices of the hidden graphs $\in \{5, 10\}$, (3) the dimensionality of the vertex features $\in \{16, 32, 64\}$ for the bio/chemo-informatics datasets and $\in \{4, 8\}$ for the social interaction datasets, (4) whether to normalize the obtained graph representations, and (5) the dropout ratio $\in \{0, 0.2\}$.

**Results.** Table 1 illustrates average prediction accuracies and standard deviations. We observe that the proposed RWNN models outperform the baselines on 2 out of the 10 datasets, while they provide the second or third best accuracy on 4 out of the remaining 8 datasets. The most successful method is the WL kernel which performs best on 4 of the 10 datasets. Furthermore, on almost all of these 4 datasets, it outperforms the other approaches with quite wide margins. Among the neural network models, the proposed RWNN models outperforms the baseline models on 5 out of the 10 datasets. On the remaining 5 datasets, GIN is the best-performing model. On the MUTAG, D&D and PROTEINS datasets, our model offers respective absolute improvements of $4.5\%$, $2.3\%$, and $1.4\%$ in accuracy over GIN. With regards to the three instances of the proposed architecture, it is quite surprising that 1-step RWNN outperforms the other two models on 5 out of the 10 datasets. It may be the case that on these datasets walks of length 2 and 3 provide no useful information for the respective classification tasks. It is true though that on most datasets, the difference in performance between the three RWNN variants is not large. Overall, the model exhibits highly competitive performance on the graph classification datasets, while the achieved accuracies follow different patterns from all the baseline methods.

## 6    Conclusion

In this paper, we presented RWNN, a novel neural network architecture for performing machine learning tasks on graphs. The proposed model generates graph representations by comparing a set of trainable "hidden graphs" against the input graphs using a variant of the well-known $P$-step random walk kernel. The conducted experiments highlight the high interpretability of the proposed model and its effectiveness on real-world datasets, where it performed comparably to state-of-the-art neural networks and kernels. As further research, we plan to extend the method to unsupervised settings, aiming at discovering the full spectrum of hidden features in the data in terms of interpretable discriminative small graphs.

## Broader Impact

GNNs have attracted a lot of attention in the past years and have been applied to a wide range of problems, mainly in chemoinformatics, bioinformatics, computer vision and natural language processing [50]. We claim that GNNs bear a stark representational power and can thus be used in more application domains. What is currently missing is interpretability of the structures learned as they are counter-intuitive. Our research in this paper offers a novel architecture tackling these problems by representing input graphs in terms of learnable discriminative small graphs that can be interpreted by human experts in the specific domain. Importantly, due to its transparency, our model can provide explanations of the results in these applications, improving understanding of decisions and of the underlying models. In addition to the intrepretability it brings, the model is also competitive in terms of performance (e. g., accuracy in classification tasks).

Depending on the application, it can help mitigate different risks or can also give rise to new opportunities. For instance, the learned graph features could assist pharmaceutical chemists in drug design or physicists in explaining laws of physics. However, there are also potential risks associated with our research. First, blind trust in our model (or machine learning models in general) which may incur risks. Second, if systems are used by individuals that do not have the necessary level of knowledge and skills, it is likely that the models will not be properly applied to the underlying problems and/or there will be an incorrect interpretation of the results. Therefore, our model, as all AI methods, needs sufficient human supervision and involvement of human experts. We thus encourage research efforts to understand the impacts and limitations of using our model in real-world scenarios.

## Acknowledgments and Disclosure of Funding

The authors would like to thank the anonymous NeurIPS reviewers for the constructive comments. We would also like to thank the NVidia corporation for the donation of a GPU as part of their GPU grant program.

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
