[Supplementary Material]

# 1  Overview

This document is supplementary material for the paper "Random Walk Graph Neural Networks". It is organized as follows. In Section 2, we define some basic concepts from linear algebra. In Section 3, we prove the Proposition 1. Section 4 provides a detailed description of the graph classification datasets. Finally, in Section 5, we give more examples of "hidden graphs" extracted from the models trained on the synthetic datasets.

# 2  Linear Algebra Concepts

In this Section, we provide definitions for concepts of linear algebra, namely the vectorization operator, the inverse vectorization operator and the Kronecker product, which we use heavily in the main paper.

**Definition 1.** *Given a real matrix $\mathbf{A} \in \mathbb{R}^{m \times n}$, the vectorization operator $vec : \mathbb{R}^{m \times n} \to \mathbb{R}^{mn}$ is defined as:*

$$vec(\mathbf{A}) = \begin{bmatrix} \mathbf{A}_{:1} \\ \mathbf{A}_{:2} \\ \vdots \\ \mathbf{A}_{:n} \end{bmatrix}$$

*where $\mathbf{A}_{:i}$ is the $i^{th}$ column of $\mathbf{A}$.*

**Definition 2.** *Given a real vector $\mathbf{b} \in \mathbb{R}^{mn}$, the inverse vectorization operator $vec^{-1} : \mathbb{R}^{nm} \to \mathbb{R}^{n \times m}$ is defined as:*

$$vec^{-1}(\mathbf{b}) = \begin{bmatrix} \mathbf{b}_1 & \mathbf{b}_{n+1} & \dots & \mathbf{b}_{n(m-1)+1} \\ \mathbf{b}_2 & \mathbf{b}_{n+2} & \dots & \mathbf{b}_{n(m-1)+2} \\ \vdots & \vdots & \vdots & \vdots \\ \mathbf{b}_n & \mathbf{b}_{2n} & \dots & \mathbf{b}_{nm} \end{bmatrix}$$

**Definition 3.** *Given real matrices $\mathbf{A} \in \mathbb{R}^{n \times m}$ and $\mathbf{B} \in \mathbb{R}^{p \times q}$, the Kronecker product $\mathbf{A} \otimes \mathbf{B} \in \mathbb{R}^{np \times mq}$ defined as:*

$$\mathbf{A} \otimes \mathbf{B} = \begin{bmatrix} \mathbf{A}_{11}\mathbf{B} & \mathbf{A}_{12}\mathbf{B} & \dots & \mathbf{A}_{1m}\mathbf{B} \\ \mathbf{A}_{21}\mathbf{B} & \mathbf{A}_{22}\mathbf{B} & \dots & \mathbf{A}_{2m}\mathbf{B} \\ \vdots & \vdots & \vdots & \vdots \\ \mathbf{A}_{n1}\mathbf{B} & \mathbf{A}_{n2}\mathbf{B} & \dots & \mathbf{A}_{nm}\mathbf{B} \end{bmatrix}$$

# 3  Proof of Proposition 1

For convenience, we restate the Proposition below.

**Proposition 1.** *Let $\mathbf{A}_1 \in \mathbb{R}^{n \times n}$ and $\mathbf{A}_2 \in \mathbb{R}^{m \times m}$ be two real matrices such that:*

$$\mathbf{A}_\times = \mathbf{A}_1 \otimes \mathbf{A}_2$$

*Then, for any $p \in \mathbb{N}$, we have that:*

$$\mathbf{A}_\times^p = (\mathbf{A}_1 \otimes \mathbf{A}_2)^p = \mathbf{A}_1^p \otimes \mathbf{A}_2^p$$

*Proof.* For $p \geq 1$, we prove the proposition by induction on $p$. It is obviously true for $p = 1$. Now take as an inductive hypothesis that it is true for some $p \geq 1$. It is well-known that the following property holds [1](Proposition 7.1.6):

$$(\mathbf{A} \otimes \mathbf{B})(\mathbf{C} \otimes \mathbf{D}) = (\mathbf{A}\,\mathbf{C} \otimes \mathbf{B}\,\mathbf{D})$$

Based on the above property, we use the induction hypothesis to obtain:

$$\mathbf{A}_\times^{p+1} = \mathbf{A}_\times^p\,\mathbf{A}_\times = (\mathbf{A}_1^p \otimes \mathbf{A}_2^p)(\mathbf{A}_1 \otimes \mathbf{A}_2) = (\mathbf{A}_1^p\,\mathbf{A}_1 \otimes \mathbf{A}_2^p\,\mathbf{A}_2) = \mathbf{A}_1^{p+1} \otimes \mathbf{A}_2^{p+1}$$

26  For $p = 0$, note that $\mathbf{A}_1^0 = \mathbf{I}_m$ and $\mathbf{A}_2^0 = \mathbf{I}_n$ where $\mathbf{I}_n$ and $\mathbf{I}_m$ are the $(m \times m)$ and $(n \times n)$ identity
27  matrices, respectively. Likewise, $\mathbf{A}_\times^0 = \mathbf{I}_{mn}$. From the definition of the Kronecker product, we have:

$$\mathbf{A}_1^0 \otimes \mathbf{A}_2^0 = \begin{bmatrix} 1\,\mathbf{A}_2^0 & 0\,\mathbf{A}_2^0 & \dots & 0\,\mathbf{A}_2^0 \\ 0\,\mathbf{A}_2^0 & 1\,\mathbf{A}_2^0 & \dots & 0\,\mathbf{A}_2^0 \\ \vdots & \vdots & \vdots & \vdots \\ 0\,\mathbf{A}_2^0 & 0\,\mathbf{A}_2^0 & \dots & 1\,\mathbf{A}_2^0 \end{bmatrix} = \begin{bmatrix} \mathbf{I}_m & 0 & \dots & 0 \\ 0 & \mathbf{I}_m & \dots & 0 \\ \vdots & \vdots & \vdots & \vdots \\ 0 & 0 & \dots & \mathbf{I}_m \end{bmatrix} = \begin{bmatrix} 1 & 0 & \dots & 0 \\ 0 & 1 & \dots & 0 \\ \vdots & \vdots & \vdots & \vdots \\ 0 & 0 & \dots & 1 \end{bmatrix} = \mathbf{I}_{nm} = \mathbf{A}_\times^0$$

28  $\square$

## 4  Real-World Graph Classification Datasets

30  We evaluated the proposed model on 10 publicly available graph classification datasets including 5
31  bio/chemo-informatics datasets: MUTAG, D&D, NCI1, PROTEINS and ENZYMES, as well as 5
32  social interaction datasets: IMDB-BINARY, IMDB-MULTI, REDDIT-BINARY, REDDIT-MULTI-
33  5K and COLLAB [5].

34  MUTAG consists of 188 mutagenic aromatic and heteroaromatic nitro compounds. The task is to
35  predict whether or not each chemical compound has mutagenic effect on the Gram-negative bacterium
36  *Salmonella typhimurium* [3]. ENZYMES contains 600 protein tertiary structures represented as graphs
37  obtained from the BRENDA enzyme database. Each enzyme is a member of one of the Enzyme
38  Commission top level enzyme classes (EC classes) and the task is to correctly assign the enzymes to
39  their classes [2]. NCI1 contains more than four thousand chemical compounds screened for activity
40  against non-small cell lung cancer and ovarian cancer cell lines [6]. PROTEINS contains proteins
41  represented as graphs where vertices are secondary structure elements and there is an edge between
42  two vertices if they are neighbors in the amino-acid sequence or in 3D space. The task is to classify
43  proteins into enzymes and non-enzymes [2]. D&D contains over a thousand protein structures. Each
44  protein is a graph whose nodes correspond to amino acids and a pair of amino acids are connected by
45  an edge if they are less than 6 Ångstroms apart. The task is to predict if a protein is an enzyme or not
46  [4]. IMDB-BINARY and IMDB-MULTI were created from IMDb, an online database of information
47  related to movies and television programs. The graphs contained in the two datasets correspond
48  to movie collaborations. The vertices of each graph represent actors/actresses and two vertices are
49  connected by an edge if the corresponding actors/actresses appear in the same movie. Each graph is
50  the ego-network of an actor/actress, and the task is to predict which genre an ego-network belongs to
51  [7]. REDDIT-BINARY and REDDIT-MULTI-5K contain graphs that model the social interactions
52  between users of Reddit. Each graph represents an online discussion thread. Specifically, each vertex
53  corresponds to a user, and two users are connected by an edge if one of them responded to at least
54  one of the other's comments. The task is to classify graphs into either communities or subreddits [7].
55  COLLAB is a scientific collaboration dataset that consists of the ego-networks of several researchers
56  from three subfields of Physics (High Energy Physics, Condensed Matter Physics and Astro Physics).
57  The task is to determine the subfield of Physics to which the ego-network of each researcher belongs
58  [7].

59  A summary of the 10 datasets is given in Table 1 below.

| Dataset | MUTAG | D&D | NCI1 | PROTEINS | ENZYMES | IMDB BINARY | IMDB MULTI | REDDIT BINARY | REDDIT MULTI-5K | COLLAB |
|---|---|---|---|---|---|---|---|---|---|---|
| Max # vertices | 28 | 5,748 | 111 | 620 | 126 | 136 | 89 | 3,782 | 3,648 | 492 |
| Min # vertices | 10 | 30 | 3 | 4 | 2 | 12 | 7 | 6 | 22 | 32 |
| Average # vertices | 17.93 | 284.32 | 29.87 | 39.05 | 32.63 | 19.77 | 13.00 | 429.61 | 508.50 | 74.49 |
| Max # edges | 33 | 14,267 | 119 | 1,049 | 149 | 1,249 | 1,467 | 4,071 | 4,783 | 40,119 |
| Min # edges | 10 | 63 | 2 | 5 | 1 | 26 | 12 | 4 | 21 | 60 |
| Average # edges | 19.79 | 715.66 | 32.30 | 72.81 | 62.14 | 96.53 | 65.93 | 497.75 | 594.87 | 2,457.34 |
| # labels | 7 | 82 | 37 | 3 | - | - | - | - | - | - |
| # attributes | - | - | - | - | 18 | - | - | - | - | - |
| # graphs | 188 | 1,178 | 4,110 | 1,113 | 600 | 1,000 | 1,500 | 2,000 | 4,999 | 5,000 |
| # classes | 2 | 2 | 2 | 2 | 6 | 2 | 3 | 2 | 5 | 3 |

Table 1: Summary of the 10 datasets that were used in our experiments.

## 5  Further Results

In this Section, we visualize four "hidden graphs" for each of the 5 synthetic datasets described in the main paper: (1) Caveman dataset, (2) Cycle dataset, (3) Grid dataset, (4) Ladder dataset, and (5) Star dataset.

### 5.1  Caveman dataset

Figure 1: Examples of "hidden graphs" extracted from the proposed model for the Caveman dataset.

### 5.2  Cycle dataset

Figure 2: Examples of "hidden graphs" extracted from the proposed model for the Cycle dataset.

 ## 5.3    Grid dataset

Figure 3: Examples of "hidden graphs" extracted from the proposed model for the Grid dataset.

 ## 5.4    Ladder dataset

Figure 4: Examples of "hidden graphs" extracted from the proposed model for the Ladder dataset.

## 5.5 Star dataset

Figure 5: Examples of "hidden graphs" extracted from the proposed model for the Star dataset.

As mentioned in the main paper, it is iteresting that the "hidden graphs" and their corresponding motifs share some similar properties.