[Reviews · NeurIPS 2020]

Review 1

Summary and Contributions: The paper proposes a novel graph neural network approach, which employs features extracted from random walk kernels to obtain graph representations in an end-to-end trainable architecture. The basic idea is to compare the graph with some “hidden graphs”, whose structure depends on the task of interest and is learned during training. The similarity is constructed with a differentiable function computing random walks of length p between the graph of interest and a number of hidden trainable graphs. The learned kernel values are used as input for the network model.

Strengths: - To get graph representations that are invariant to node permutation concepts from graph kernels, and in particular random walk kernels, are exploited. - Efficient way to compute the kernel and matrix power based on sparse multiplication. However, these ideas are generally implicit in the message passing framework. - An extension of the classical random walk kernel is proposed to account for continuous attributes, by computing their similarities across the nodes

Weaknesses: - In the synthetic experiments the interpretation and similarity of the retrieved motifs is not clearly quantified. Possibly, an evaluation procedure to establish the level of interpretability based on an explicit counting metric would be beneficial. - I’d recommend comparing RWNN to the random walk kernel and in particular to the k-step random walk kernel [36], given the close resemblance with the proposed method. - The results are competitive but not outstanding. Possibly, using large graph datasets would show the benefit of this hybrid architecture, as compared to either the kernel and network based methods.

Correctness: As far as I can see, the method and experimental setting are correct.

Clarity: I find the paper clearly written and easy to read.

Relation to Prior Work: The differences and similarities to existing approaches seem to be clearly defined and are summarised in a specific section.

Reproducibility: Yes

Additional Feedback: Thank you for your answer. I updated my score accordingly.


Review 2

Summary and Contributions: The paper proposes a new kind of graph processing layer based on the application of a differentiable version of the Random Walk kernel computed between the graph and prototype graphs that are learned via backpropagation.

Strengths: -The proposed idea is interesting and different from many proposals in literature -The paper addresses the interesting issue of merging graph kernels and graph neural networks -This should not be a strength of a paper in principle, but the experimental evaluation procedure is correct, and this is not so common these days.

Weaknesses: -In my understanding, the learned (weighted) graphs can be in principle fully connected, so not necessarily interpretable.

Correctness: The claims seem correct and the experimental evaluation is correct.

Clarity: yes

Relation to Prior Work: yes

Reproducibility: Yes

Additional Feedback: -The learned graphs are weighted (positive weights). They can thus be fully connected in principle. If I understood correctly, this fact may negatively impact the interpretability of the learned model, on which you stress quire a bit in the writing and also in the broader impact statement. In the graphs reported in the supplementary material, did you use a threshold value to represent edges? I suggest to discuss this drawback in the paper, and to possibly analyse cases in which the learned graphs are interpretable and when they are not. Minor remarks: -l 47: also graph kernels map the input graphs in vectorial spaces. The main difference is the dimensionality of the space. I suggest to elaborate more on this point. -l58-67 "better or comparable to the state of the art". On NCI1 it is not true. I think the contribution of the paper is pretty interesting, no need to oversell the results. In this paper, it is the novel approach to graph processing in neural networks that is interesting. -l87: graph neural networks were introduced before Scarselli in: A. Sperduti, A. Starita, Supervised neural networks for the classification of structures. IEEE Trans. Neural Networks 8, 714–735 (1997). and basically at the same time in: A. Micheli, Neural network for graphs: A contextual constructive approach. IEEE Trans. Neural Networks 20, 498–511 (2009). ---- I confirm my score after the rebuttal and discussion phase.


Review 3

Summary and Contributions: The authors propose a novel neural network model for graph data based on the random walk kernel.

Strengths: The approach is interesting and novel. The method is sound.

Weaknesses: Comparisons with competitive explainability methods are lacking.

Correctness: The methodology is correct.

Clarity: The paper is well written.

Relation to Prior Work: Comparisons with competitive explainability methods are lacking.

Reproducibility: Yes

Additional Feedback: The authors propose a novel neural network model for graph data based on the random walk kernel. The proposed approach is interesting to me. My concerns lie in the utility of the proposed method. The proposed method does not necessarily outperform the existing methods in many tasks. Although its interpretability is preferable, other graph kernels also have interpretability, and there are interpretation methods for graph neural networks such as GNNExplainer [Ying et al. 2019]. To show the effectiveness of the interpretability of the proposed method, it is necessary to compare the proposed method with such interpretation methods, discuss the differences, and analyze the hidden graphs in the real-world datasets extensively. Overall, more adequate experiments and discussions for confirming the utility of the proposed method are needed. Reference Rex Ying, Dylan Bourgeois, Jiaxuan You, Marinka Zitnik, Jure Leskovec. GNNExplainer: Generating Explanations for Graph Neural Networks. NeurIPS 2019. Federico Baldassarre, Hossein Azizpour. Explainability Techniques for Graph Convolutional Networks. ICML 2019 workshop. Hao Yuan, Jiliang Tang, Xia Hu, Shuiwang Ji. XGNN: Towards Model-Level Explanations of Graph Neural Networks. KDD 2020. ======= After rebuttal: Thank you for the response. That convinced me about the novelty of the proposed approach. Hence I raised my score. I'd still like to see the interpretability of the proposed method in real-world datasets if possible.


Review 4

Summary and Contributions: Authors provide an alternative to Graph Neural Networks (GNNs) or Message Passing Neural Nets (MPNN) for graph-level classification task. Specifically, they use a learnable graph kernel. The kernel uses "hidden graphs" (updated through training): their adjacency matrices is continuous, restricted to be positive and symmetric. This gives a competitive and interpretable model for graph classification.

Strengths: * Show a way of learning kernels on graphs combined with a neural network. The learned kernels are interpretable, as opposed to multi-layer Graph Neural Networks (GNNs) * The first presentation looks O((n1 n2) ^ 2) to compute a forward pass, but they describe an O(n1 n2) algorithm for doing the computation; with n1 and n2 being the dimensions of the kernel. * They mention interpretability, but they could bring some chemistry (as their paper uses such datasets) aspect why interpretability could be useful for chemists, or other fields they want to highlight.

Weaknesses: * The claim that MPNNs will treat nodes as "sets", ignoring edge information, is too strong: it is easy to show cases where the author's statement is not true. Please tone-down this argument because the reader will get shocked. If you want to keep it, provide theorems and proofs detailing the classes of graphs and MPNNs that are, as you say, ignorant about the edges. The claim can be corrected by limiting to the graph-pooling (as they called it, "readout") that go from node to graph representations. * Why not have experiments with different sizes of hidden graphs, as shown by the figure?

Correctness: Paper seems correct to me: Math and text [except the too-strong of a tone argument, in weaknesses]. I didnt spot many spelling / grammer mistakes, though the last sentence of the conclusion ...

Clarity: The paper is well written and easy to follow. They take the time to explain concepts from scratch making the paper stand on its own for the conference audience.

Relation to Prior Work: Related work is described well. Would it make sense to compare against https://arxiv.org/abs/1904.09671 or at least mention it in the write-up if it is easy to highlight any similarities/differences?

Reproducibility: Yes

Additional Feedback:

[Author Response · NeurIPS 2020]

We would like to thank the reviewers for their constructive comments. Below, we try to respond to their main comments.

R1: **Quantitative evaluation**. Indeed, a quantitative evaluation would improve the significance of the empirical results. Note that each motif exhibits some distinct properties and can be considered as a graph-feature. For instance, a star graph contains a single node with high degree. The caveman graphs contain many triangles, the ladder graphs consist of cycles of size 4, etc. For each dataset, we plan to measure how much these properties are satisfied by the learned hidden graphs and we will report the results in the revised manuscript.

R1: **Comparing against $k$-step RW**. We have started evaluating the $k$-step RW kernel (the implementation contained in the graphkernels package) on the 10 datasets. We obtained the following average accuracies on MUTAG, ENZYMES and NCI1: 72.39 ($\pm$ 6.9), 19.00 ($\pm$ 4.4) and 54.06 ($\pm$ 1.6), respectively. We observed that computing the kernel matrix on the larger datasets (DD, REDDIT-BINARY, REDDIT-MULTI-5K, COLLAB) takes more than 1 day or requires large amounts of memory (more than 16GB of RAM). Thus, it seems that the $k$-step RW kernel is fairly weak and suffers from time/memory issues. With this in mind, we are not sure if it is worth adding this baseline to the paper.

R1: **Large graph datasets**. This is definitely on our agenda for future work. We plan to evaluate the proposed architecture on the QM9 dataset which contains more than 100k samples.

R2: **Fully connected learned graphs**. In our implementation, a hidden graph of order $n$ is associated with $n(n-1)/2$ trainable parameters. We do not directly treat these values as the weights of the edges between the different pairs of nodes, but we first apply the ReLU activation function. Therefore, all the negative values are set equal to 0, and the corresponding edges are essentially removed. Even though the learned graphs can be complete, we empirically observed that in most cases, they are fairly sparse. We will make this clear in the revised manuscript.

R2: **Feature space of kernels vs. that of proposed model**. Indeed, a graph kernel maps graphs to some Hilbert space where each dimension typically corresponds to some substructure (e.g., shortest path of specific length, a specific subtree, etc). This space is different from the one to which our model maps graphs. In our case, each dimension corresponds to the "similarity" of the input graph and some hidden graph. Furthermore, since the proposed model is end-to-end trainable, this space is not fixed, but it depends on the structure of the hidden graphs.

R2, R3: **Empirical performance**. It is true that the empirical results are not very impressive in general. Even though the proposed model does not provide a new state-of-the-art in graph classification, still it outperforms the majority of the GNN baselines on most datasets. We agree that the main strength of the paper is in the novelty of the proposed architecture, not in the empirical performance (though we believe that it is not that bad).

R2: **Papers that first introduced GNNs**. We thank the reviewer for pointing us to these papers. We will rephrase the sentence and cite the above papers.

R3: **Comparing against explainability techniques for GNNs**. We should first mention that our paper is different from these works in that the main focus of the proposed model is not on providing interpretable explanations for its predictions, but on dealing with graph-level supervised learning tasks. Note also that the methods proposed in these 3 papers are mainly applied as a post-processing step: they take a trained MPNN and its predictions, and they return an explanation of these predictions. On the other hand, in the case of the proposed model, these explanations come as a byproduct of the learning process. Comparing against these methods seems thus to be out of the scope of this paper.

Furthermore, note that the explanations generated by the proposed model are similar to those of the method presented in the third paper (both belong to the family of model-level methods). Note that the third paper has not been published yet and was posted on arXiv 2 days before the submission deadline of NeurIPS. Therefore, comparing against this method was also practically infeasible.

R4: **MPNNs ignore edge information**. We agree with the reviewer that it is graph-pooling that treats graphs as sets/multisets of node representations. Since the graph structure (e.g., subtree patterns) is encoded into these representations, MPNNs (message passing along with graph-pooling) do not ignore edge information, but they take it into account. We will rephrase our claim as suggested by the reviewer.

R4: **Experiments with different sizes of hidden graphs**. This is mainly due to computational reasons. In our implementation, the parameters of all hidden graphs are combined into a single trainable matrix. This allows us to perform operations such as batch matrix multiplications that benefit greatly from GPUs. To employ hidden graphs of different sizes, we would need to have more than one trainable matrices, and that would slightly increase the running time of the model.

R4: **Comparing against DDGK**. This paper is indeed related to our work. One major difference is that our model is supervised, while DDGK is unsupervised. We will discuss the difference between this approach and our work in the related work section, as suggested. The source code of DDGK is also publicly available, and we have started evaluating it on the 10 graph classification datasets. We will report the results in the revision.

[Meta-Review · NeurIPS 2020]

This paper presents a nice new GNN architecture based on random walk kernels. All reviewers unanimously support accepting this paper. I hope in the final version the authors can improve the presentation and clear the clarity issues raised in the rebuttal process and also discuss parallel related work that you didn’t have chance to in this submission.